# Prognostic Effect of the Dose of Radiation Therapy and Extent of Lymphadenectomy in Patients Receiving Neoadjuvant Chemoradiotherapy for Esophageal Squamous Carcinoma

**DOI:** 10.3390/jcm11175059

**Published:** 2022-08-28

**Authors:** Chu-Pin Pai, Ling-I Chien, Chien-Sheng Huang, Han-Shui Hsu, Po-Kuei Hsu

**Affiliations:** 1Division of Thoracic Surgery, Department of Surgery, Taipei Veterans General Hospital, Taipei 112201, Taiwan; 2Division of Thoracic Surgery, Department of Surgery, Lotung Poh-Ai Hospital, Yilan 256, Taiwan; 3Department of Nursing, Taipei Veterans General Hospital, Taipei 112201, Taiwan; 4School of Medicine, National Yang Ming Chiao Tung University, Taipei 112304, Taiwan

**Keywords:** esophageal cancer, esophageal squamous cell carcinoma, neoadjuvant, chemoradiotherapy, radiation dose, lymph node dissection

## Abstract

Neoadjuvant chemoradiotherapy has been used for patients with locally advanced esophageal squamous cell carcinoma (ESCC). However, the optimal dose of radiation therapy and the effect of lymphadenectomy after neoadjuvant therapy on patient outcomes are uncertain. We retrospectively reviewed the data of patients who received neoadjuvant therapy followed by surgery for ESCC. Overall survival (OS), disease-free survival (DFS), and perioperative outcomes were compared between patients who received radiation doses of 45.0 Gy (PF4500) and 50.4 Gy (PF5040). Subgroup analysis was performed based on the number of lymph nodes removed through lymph node dissection (LND). Data from a total of 126 patients were analyzed. No significant differences were found in 3-year OS and DFS between the PF4500 and PF5040 groups (OS: 45% versus 54%, *p* = 0.218; DFS: 34% versus 37%, *p* = 0.506). In both groups, no significant differences were found in 3-year locoregional-specific DFS between patients with a total LND number ≤17 and >17 (PF4500, 35% versus 50%, *p* = 0.291; PF5040 group, 45% versus 46%, *p* = 0.866). The PF5040 and PF4500 groups were comparable in terms of survival outcomes and local control. Although no additional survival benefits were identified, the extent of LND should not be altered according to the radiation dose.

## 1. Introduction

Esophageal squamous cell carcinoma (ESCC) is one of the most aggressive malignancies. A multimodal approach including adequate radiotherapy (RT) for local control with sensitizing chemotherapy followed by surgery has become the mainstay treatment for patients with locally advanced ESCC [1,2,3,4]. Carboplatin and paclitaxel with concurrent 41.4 Gy RT (CROSS) is the current standard of care for neoadjuvant chemoradiotherapy (NCRT) in Western regions, mainly in response to esophageal adenocarcinoma [5]. In Asia, more than 90% of all esophageal cancer diagnosed is ESCC. Cisplatin plus 5-fluorouracil (PF)-based regimens with an RT dose between 41.4 and 50.4 Gy has proven effective in previous clinical trials and is frequently used in Asia [5,6,7,8]. Despite the wide range of acceptable RT doses, the doses most commonly used in Taiwan are 45.0 and 50.4 Gy [9].

Extensive lymph node dissection (LND) during surgery is also used in addition to RT to improve local control [10,11,12,13]. However, previous studies have obtained conflicting results in terms of its survival benefits in patients who have received neoadjuvant chemoradiotherapy (NCRT) [13,14,15,16]. Whether the differences in outcomes are related to variations in RT doses and whether the improved local control derived from RT mitigates the effect of LND remains unclear. 

To address these knowledge gaps, we conducted the present study to assess the outcomes of two groups, one receiving PF with an RT dose of 45.0 Gy (PF4500) and the other receiving PF with an RT dose of 50.4 Gy (PF5040). We then examined the effects of LND in each group. The overall survival (OS), disease-free survival (DFS), pathological complete response (pCR), local control, and perioperative outcomes were compared.

## 2. Materials and Methods

The study protocol was reviewed and approved by the Institutional Review Board of Taipei Veterans General Hospital (TPEVGH2015-06-001BC). 

### 2.1. Study Population 

We retrospectively reviewed the clinical charts of patients diagnosed with esophageal malignancies at Taipei Veterans General Hospital between January 2008 and December 2018. The inclusion criteria were patients with SCC who received NCRT with a planned radiation dose of 45.0 to 50.4 Gy followed by surgery. The exclusion criteria were cervical esophageal tumors and other malignancies diagnosed synchronously with or within 5 years prior to esophageal cancer. 

The staging workup included a systemic physical examination, standard laboratory screening, esophagogastroscopy (endoscopic ultrasound), bronchoscopy for tumors in the upper or middle third of the esophagus, computed tomography (CT) scanning from the neck to the upper abdomen, and whole-body fluorodeoxyglucose positron emission tomography/CT. A multidisciplinary team conference was held, comprising surgeons, medical and radiation oncologists, gastroenterologists, pathologists, radiologists, and special nurses, for discussion and recommendations regarding treatment planning and the modification thereof.

### 2.2. NCRT and Surgery

NCRT consisted of 80 mg/m^2^ of cisplatin administered intravenously on Day 1, followed by continuous intravenous infusion of 600 mg/m^2^ of 5-fluorouracil administered from Days 1 to 4, at 4-week intervals for two courses of chemotherapy, with concurrent external beam RT (cumulative dose of 45.0 or 50.4 Gy, in fractions of 1.8–2.0 Gy). The decision of utilizing a RT of 45.0 or 50.4 Gy was made based on the updated evidences at the time of diagnosis, disease status, and medical condition of the patient. To optimize the treatment planning, a thorough discussion in the multidisciplinary team conference was performed. The clinical target volume was defined as the gross target tumor volume delineated on CT scans and other diagnostic images, along with 3–5 cm and 5 cm cephalad and caudal margins, respectively. Toxicity was graded according to the Common Terminology Criteria for Adverse Events (CTCAE v5.0). Restaging workup was arranged approximately 4 weeks after the completion of RT, and the interval between the end of RT and esophagectomy was approximately 6–8 weeks. In our institution, the thoracic stage involved thoracotomy or video-assisted thoracoscopic surgery for esophagectomy and LDN. The abdominal stage involved either laparotomy or laparoscopic surgery. Esophagogastric anastomosis was performed in the chest with the stapling method or at the neck with either the stapling or hand-sewn technique. Radical cervical LND was performed by an otolaryngologist if metastasis was clinically suspected. Postoperative complications were defined according to the guidelines of the Esophagectomy Complications Consensus Group [17].

Pathological staging was determined according to the American Joint Committee on Cancer staging criteria, eighth edition. All surgical specimens were assessed by pathologists for pathological response to NCRT, and the primary tumor was graded with a modified Ryan score [18]. Specifically, a score of 0 indicates complete response (no viable cancer cells), 1 refers to near-complete response (single cells or rare small groups of cancer cells), 2 indicates partial response (residual cancer with evident tumor regression, but more than single cells or rare small groups of cancer cells), and 3 indicates poor or no response (extensive residual cancer with no evident tumor regression). The pathological stages after neoadjuvant therapy were ypT and ypN. Downstaging of T or N categories was defined as ypT or ypN stages lower than the clinical T or N stage, respectively. Pathological complete response was defined as no residual cancer cells and no lymph node metastasis (ypT0N0). After the operation, follow-up evaluations were arranged including clinical and laboratory testing, and chest CT every 3–4 months for the first 2 years, every 6 months between the second and fifth years, and every year after the fifth year. Disease recurrence was classified as locoregional recurrence (LRR) or distant metastasis. LRR included recurrence at the anastomotic site, in the area of previous resection, or within the region of cervical, mediastinal, or upper abdominal LND. Distant metastasis was defined as distant lymph node dissemination, hematogenous metastasis to solid organs, or recurrence in the peritoneal or pleural cavities.

### 2.3. Statistics

Continuous variables were either recorded as means and compared using Student’s *t* test or were summarized as medians and compared using the Mann–Whitney *U* test. Categorical variables were recorded as absolute counts and compared using the chi-square test or Fisher’s exact test. OS was defined as the time from the beginning of RT until death or the last known follow-up, based on either medical records or a follow-up phone call. Survival curves were plotted using the Kaplan–Meier method and compared using the log-rank test. Univariable and multivariable Cox regression modeling was used to identify prognostic factors. Factors with a *p* value < 0.05 in univariable analysis were included in multivariable modeling. All statistical analyses were conducted using the Statistical Product and Service Solutions (SPSS, version 25; IBM Corp, Armonk, NY, USA), and a two-sided *p* value < 0.05 was considered statistically significant.

## 3. Results

### 3.1. Clinicopathological Characteristics of the Study Patients

During the study period, a total of 145 patients met the criteria. After the exclusion of patients with cervical SCC (*n* = 5) and synchronous malignancy (*n* = 14), the data of the remaining 126 patients were analyzed. The clinical and pathological characteristics of the patients are summarized in Table 1. Among the 126 patients, 41 received a radiation dose of 45.0 Gy (PF4500 group), and the remaining 85 received 50.4 Gy (PF5040 group). More male patients were included in the PF4500 group than in the PF5040 group (97.6% versus 84.7%, *p* = 0.035); otherwise, no significant differences were found in the baseline factors. The surgical parameters, which included resection radicalness, number of lymph nodes removed through LND, and grade of tumor or nodal regression were not statistically different between the two groups.

### 3.2. Tumor Response

In the PF4500 and PF5040 groups, 13 (31.7%) and 39 (45.9%) patients exhibited pathological complete response (pCR), respectively (*p* = 0.176). The rate of ypT0 (34.1% vs. 49.4%), ypN0 (70.7% vs. 78.8%), T (68.3% vs. 72.9%), and N (73.2% vs. 77.6%) downstaging was similar between the PF4500 and PF5040 groups.

### 3.3. Survival, Prognostic Factors, and Local Control

In the survival analysis, the median follow-up time for all patients was 22.6 (interquartile range: 11.9–42.5) months. The 3-year OS and DFS rates in the entire cohort were 52% and 36%, respectively, and the 3-year locoregional-specific DFS rate was 46%. No significant differences were found in 3-year OS and DFS between patients in the PF4500 and PF5040 groups (OS: 45% versus 54%, *p* = 0.218; DFS: 34% versus 37%, *p* = 0.506, Figure 1a,b). Further analysis based on the total LND number revealed no significant differences in 3-year OS and DFS (Figure 2a,b). 

Regarding 3-year locoregional-specific DFS, no significant differences were found between patients in the PF4500 and PF5040 groups (3-year locoregional DFS: 42% versus 47%, *p* = 0.518, Figure 3a). Subgroup analysis of local control was performed according to LND number. As shown in Figure 3b, in both the PF4500 and PF5040 groups, no significant differences were found in 3-year locoregional-specific DFS between patients with a total LND number ≤ 17 and >17 (PF4500 group, 35% versus 50%, *p* = 0.291; PF5040 group, 45% versus 46%, *p* = 0.866). Between the PF4500 and PF5040 groups, the patients with a total LND number ≤ 17 had comparable 3-year locoregional-specific DFS (50% versus 46%, *p* = 0.788). Between the PF4500 and PF5040 groups, the patients with a total LND number > 17 also had comparable 3-year locoregional-specific DFS (35% versus 45%, *p* = 0.336).

A Cox proportional hazards regression model was used to analyze prognostic factors for the OS (Table 2) and DFS (Table 3) of the entire cohort. The significant prognostic factors in univariable analysis for OS included resection radicalness, pCR, ypT0, T downstaging, tumor differentiation, perineural invasion (PNI), lymphovascular invasion (LVI), and pathological stage. Among these factors, pCR (hazard ratio [HR]: 0.44, 95% confidence interval [CI]: 0.24–0.82, *p* = 0.010) remained an independent prognostic factor in the multivariable analysis. The significant prognostic factors in univariable analysis for DFS included resection radicalness, pCR, ypT0, T downstaging, tumor differentiation, PNI, LVI, and pathological stage. Among these factors, pCR (HR: 0.56, 95% CI: 0.32–0.96, *p* = 0.035) and pathological stage (HR: 2.65, 95% CI: 1.11–6.33, *p* = 0.029) remained independent prognostic factors in the multivariable analysis. 

Regarding locoregional-specific DFS, a Cox proportional hazards regression model was used to identify prognostic factors (Table 4). The significant prognostic factors in univariable analysis for locoregional recurrence included resection radicalness, pCR, ypT0, T downstaging, tumor differentiation, PNI, and LVI. Among these factors, pCR (HR: 0.53, 95% CI: 0.30–0.92, *p* = 0.023) remained an independent prognostic factor in the multivariable analysis.

### 3.4. Perioperative Outcomes

The details of the perioperative course are listed in Table 5. The duration of operative time was longer in the PF4500 group than in the PF5040 group. No difference was found in the length of hospital stay between the two groups. Regarding the postoperative course, vocal cord palsy was found in 16.7% of the entire cohort of patients, 47.6% of which were the transient type and recovered spontaneously. Moreover, the rates of chyle leaks, anastomotic leaks, pneumonia, wound infection, overall complication grading higher than III, and short-term postoperative mortality were similar between the PF4500 and PF5040 groups. The adverse events of NCRT in the two groups were summarized in Table 6. Comparable results were identified in all grade and grade ≥ 3 adverse events. No severe adverse events intercepting the neoadjuvant therapy were reported.

## 4. Discussion

The promising survival outcome of neoadjuvant therapy followed by surgery has been well demonstrated compared to surgery alone in previous trials, and the use of NCRT has become the standard of care for patients with resectable esophageal SCC [19]. An RT dose between 41.4 and 50.4 Gy has proven effective and is frequently used; however, the ideal radiation dose remains unidentified [20,21,22,23]. Hypothetically, a higher RT dose improves local control and results in a better survival outcome. Hence, numerous studies have focused on comparing the effects of different RT doses, but with varying results [7,24]. In a pooled analysis by Li et al., the OS rates of a 50.0–50.4 Gy group were significantly higher than those in a 44.0–46.0 Gy group [23]. Conversely, no added survival benefits of a higher RT dose were reported by others [25,26,27,28], which is compatible with our finding. 

Despite the inconclusive survival benefits of different radiation doses, some studies have identified an association between a higher RT dose and a higher likelihood of pCR, but this did not translate into better survival [26,29]. In addition, other studies have demonstrated no RT dose-escalation effect on the pCR or T- and N- downstaging rate [30,31,32]. In our study, comparable rates of pCR (45.9% versus 31.7%, *p* = 0.176), ypT0 (49.4% versus 34.1%, *p* = 0.106), and ypN0 (78.8% versus 70.7%, *p* = 0.318) were found between the PF5040 and PF4500 groups. We also identified a similar T- and N- downstaging rate between the two groups. Current evidence is inconclusive as to whether better tumor response or improved local control can be achieved by a higher RT dose. 

NCRT exerted adverse effects as well as survival benefits. In a multi-center study that used 45.0 Gy in NCRT, higher rates of chylothorax were found in the NCRT group compared with an upfront surgery group [33]. Therefore, caution should be taken when using higher RT doses in NCRT. The safety of different RT doses has been investigated in previous studies. For example, the INT 0123 trial [7], which mostly comprised SCC patients (86%), compared the outcome and toxicity of high-dose (64.8 Gy) and standard-dose (50.4 Gy) RT in NCRT. An increased incidence of treatment-related deaths was identified in the high-dose arm, leading to only 67% of patients receiving radiation according to the protocol compared with 83% in the standard-dose arm. On the other hand, in a study of the National Cancer Database by Ising et al., comparable rates of R0, 30-day, and 90-day mortality were identified between low-dose (41.4 Gy) and high-dose (50.0 or 50.4 Gy) groups after propensity score matching [31]. The postoperative length of stay and unplanned readmission within 30 days, which can serve as a surrogate for post-operative complications, were also similar between the groups. In our study, similar rates of neoadjuvant therapy related adverse events were identified between the PF4500 and PF5040 groups. Regarding the postoperative course, rates of overall complications with Clavien–Dindo grade ≥ III, postoperative length of stay, unplanned readmission within 30 days and short-term mortality were also comparable between the two groups. The PF5040 group, despite using a higher RT dose, exhibited similar results of treatment toxicity compared to the PF4500 group. Overall, both regimens in our study should be considered as a safe and effective neoadjuvant treatment.

Both LND and RT are aimed at improving local control and achieving tumor downstaging, which raises the question of whether extended LND plays the same role in patients who receive neoadjuvant therapy. Both Visser et al. and Guo et al. reported that a higher LND number was associated with improved outcomes and suggested that radical lymphadenectomy should be considered the standard of care after NCRT at doses of 40.0 or 41.4 Gy [34,35]. By contrast, in the work of Talsma et al. and Shridhar et al., which involved doses of 41.4 Gy and 50.40 Gy, benefits of lymphadenectomy were not observed [13,15]. In our study, when stratified by total LND number, the OS, DFS and locoregional-specific DFS were comparable between the PF4500 and PF5040 groups (Figure 2a,b and Figure 3b). Regarding loco-regional control, a more aggressive LND did not translate into a significantly improved outcome. On the other hand, despite using a higher RT dose in the PF5040 group, the ypN0 and ypN downstaging rate was similar comparing to the PF4500 group. In terms of loco-regional control of lymph node metastasis, no significant superiority was observed in the PF5040 group over the PF4500 group, and a disfavoring outcome could occur if the LND was compromised. Therefore, despite no additional benefits being detected, LND should not be altered between the two regimens. 

Certain limitations of the present study should be acknowledged. First, in the setting of neoadjuvant therapy, the most common radiation dose used in our hospital is 45.0 Gy or 50.4 Gy, and therefore we were unable to compare outcomes with patients who received lower doses such as 41.4 Gy. Second, because of its retrospective design, the decision between PF4500 and PF5040 was not randomized but rather based on the decision of the multidisciplinary team in our hospital. Therefore, we could not completely exclude an inherent selection bias. Furthermore, a relatively small number of patients was included after each group was divided based on LND number.

## 5. Conclusions

We observed comparable results of OS, DFS, and locoregional control between the PF4500 and PF5040 groups. No added benefit of a higher LND number was identified in either group. Based on the findings of this retrospective study, the PF5040 group exhibited no superiority compared with the PF4500 group in terms of local control; therefore, LND should not be altered. Higher radiation doses during neoadjuvant therapy should not be used as a surrogate for suboptimal lymphadenectomy.

## Figures and Tables

**Figure 1 jcm-11-05059-f001:**
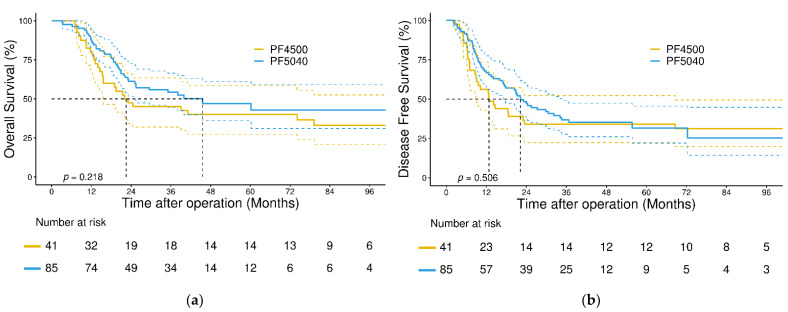
(**a**) Kaplan–Meier analysis of overall survival in the entire cohort, stratified according to regimens; (**b**) Kaplan–Meier analysis of disease-free survival in the entire cohort, stratified according to regimens.

**Figure 2 jcm-11-05059-f002:**
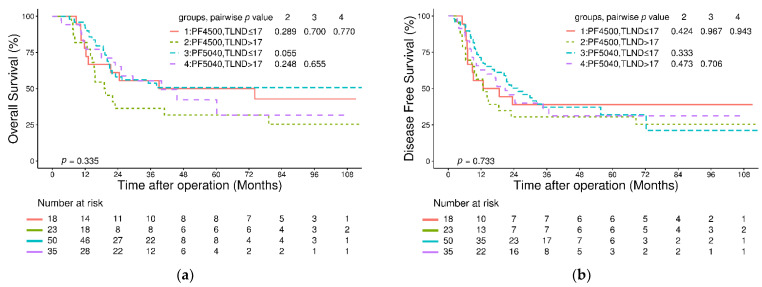
(**a**) Kaplan–Meier analysis of overall survival in the entire cohort stratified according to regimens and total LND numbers; (**b**) Kaplan–Meier analysis of disease-free survival in the entire cohort stratified according to regimens and total LND numbers. Abbreviations: TLND: total lymph node dissection.

**Figure 3 jcm-11-05059-f003:**
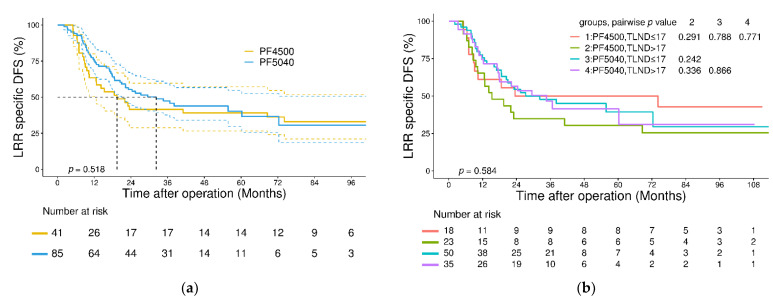
(**a**) Kaplan–Meier analysis of LRR specific disease-free survival in the entire cohort stratified according to regimens; (**b**) Kaplan–Meier analysis of LRR specific disease-free survival in the entire cohort stratified according to regimens and total LND numbers. Abbreviations: LRR: locoregional recurrence; DFS: disease-free survival; TLND: total lymph node dissection.

**Table 1 jcm-11-05059-t001:** Clinical characteristics of patients categorized by total LND number in the PF4500 and PF5040 groups.

	Total N = 126	PF4500 N = 41					PF5040 N = 85			
		*p* *	Total	TLND		*p* **	Total	TLND		*p* ***
				≤17 N = 18	>17 N = 23			≤17 N = 51	>17 N = 34	
Age (years; median, IQR)	57.0 (50.0–63.3)	0.284	56.0 (47.0–63.0)	58.0 (43.0–64.0)	55.0 (48.0–60.0)	0.703	57.0 (50.0–64.0)	57.0 (50.0–63.3)	59.0 (50.0–64.0)	0.425
Sex (%)		0.035				0.370				0.549
Female	14 (11.1)		1 (2.4)	0 (0)	1 (4.3)		13 (15.3)	9 (17.6)	4 (11.8)	
Male	112 (88.9)		40 (97.6)	18 (100)	22 (95.7)		72 (84.7)	42 (82.4)	30 (88.2)	
Tumor location (%)		0.234				0.856				0.450
Proximal	28 (22.2)		13 (31.7)	5 (27.8)	8 (34.8)		15 (17.6)	8 (15.7)	7 (20.6)	
Middle	57 (45.2)		16 (39.0)	8 (44.4)	8 (34.8)		41 (48.2)	23 (45.1)	18 (52.9)	
Distal	41 (32.5)		12 (29.3)	5 (27.8)	7 (30.4)		29 (34.1)	20 (39.2)	9 (26.5)	
Tumor length (cm; median, IQR)	5.0 (4.0–7.0)	0.504	5.0 (4.0–8.0)	5.0 (4.0–7.0)	5.5 (4.0–10.0)	0.333	5.0 (4.0–7.0)	5.0 (4.0–7.2)	5.0 (4.0–6.0)	0.275
cT stage (%)		0.130				<0.001				0.094
1	10 (7.9)		4 (9.8)	1 (5.6)	3 (13.0)		6 (7.1)	2 (3.9)	4 (11.8)	
2	16 (12.7)		9 (22.0)	9 (50.0)	0 (0)		7 (8.2)	2 (3.9)	5 (14.7)	
3	97 (77.0)		27 (65.9)	8 (44.4)	19 (82.6)		70 (82.4)	46 (90.2)	24 (70.6)	
4	3 (2.4)		1 (2.4)	0 (0)	1 (4.3)		2 (2.4)	1 (2.0)	1 (2.9)	
cN stage (%)		0.988				0.339				0.611
0	13 (10.3)		4 (9.8)	2 (11.1)	2 (8.7)		9 (10.6)	6 (11.8)	3 (8.8)	
1	66 (52.4)		21 (51.2)	11 (61.1)	10 (43.5)		45 (52.9)	24 (47.1)	21 (61.8)	
2	39 (31.0)		13 (31.7)	3 (16.7)	10 (43.5)		26 (30.6)	18 (35.3)	8 (23.5)	
3	8 (6.3)		3 (7.3)	2 (11.1)	1 (4.3)		5 (5.9)	3 (5.9)	2 (5.9)	
Total LND number (median, IQR)	17.0 (11.0–23.0)	0.210	19.0 (12.5–24.0)	11.5 (8.5–15.3)	23.0 (19.0–28.0)	<0.001	16.0 (10.5–22.0)	11.0 (9.0–14.0)	24.0 (20.0–36.0)	<0.001
yp T (%)		0.386				0.875				0.457
0	56 (44.4)		14 (34.1)	7 (38.9)	7 (30.4)		42 (49.4)	24 (47.1)	18 (52.9)	
1	17 (13.5)		7 (17.1)	4 (22.2)	3 (13.0)		10 (11.8)	5 (9.8)	5 (14.7)	
2	21 (16.7)		9 (22.0)	3 (16.7)	6 (26.1)		12 (14.1)	6 (11.8)	6 (17.8)	
3	26 (20.6)		8 (19.5)	3 (16.7)	5 (21.7)		18 (21.2)	14 (27.5)	4 (11.8)	
4	6 (4.8)		3 (7.3)	1 (5.6)	2 (8.7)		3 (3.5)	2 (3.9)	1 (2.9)	
yp N (%)		0.243				0.477				0.921
0	96 (76.2)		29 (70.7)	14 (77.8)	15 (65.2)		67 (78.8)	40 (78.4)	27 (79.4)	
1	22 (17.5)		8 (19.5)	4 (22.2)	4 (17.4)		14 (16.5)	9 (17.6)	5 (14.7)	
2	6 (4.8)		2 (4.9)	0 (0)	2 (8.7)		4 (4.7)	2 (3.9)	2 (5.9)	
3	2 (1.6)		2 (4.9)	0 (0)	2 (8.7)		0 (0)	0 (0)	0 (0)	
T downstaging (%)	90 (71.4)	0.588	28 (68.3)	13 (72.2)	15 (65.2)	0.632	62 (72.9)	35 (68.6)	27 (79.4)	0.273
N downstaging (%)	96 (76.2)	0.580	30 (73.2)	13 (72.2)	17 (73.9)	0.903	66 (77.6)	41 (80.4)	25 (73.5)	0.596
Differentiation (%)		0.213				0.677				0.944
No residual tumor	55 (43.7)		13 (31.7)	6 (33.3)	7 (30.4)		42 (49.4)	24 (47.1)	18 (52.9)	
Well	3 (2.4)		1 (2.4)	1 (5.6)	0		2 (2.4)	1 (2.0)	1 (2.9)	
Moderately	46 (36.5)		16 (39.0)	8 (44.4)	8 (34.8)		30 (35.3)	18 (35.3)	12 (35.3)	
Poorly	13 (10.3)		6 (14.6)	2 (11.1)	4 (17.4)		7 (8.2)	5 (9.8)	2 (5.9)	
Unknown	9 (7.1)		5 (12.2)	1 (5.6)	4 (17.4)		4 (4.7)	3 (5.9)	1 (2.9)	
PNI (%)		0.449				0.871				0.227
Negative	107 (86.3)		35 (89.7)	16 (88.9)	19 (90.5)		72 (84.7)	41 (80.4)	31 (91.2)	
Positive	17 (13.7)		4 (10.3)	2 (11.1)	2 (9.5)		13 (15.3)	10 (19.6)	3 (8.8)	
LVI (%)		0.520				0.970				0.384
Negative	101 (81.0)		31 (77.5)	14 (77.8)	17 (77.3)		70 (82.4)	40 (78.4)	30 (88.2)	
Positive	24 (19.0)		9 (22.5)	4 (22.2)	5 (22.7)		15 (17.6)	11 (21.6)	4 (11.8)	
TRG (%)		0.085				0.727				0.287
0	54 (42.9)		14 (34.1)	7 (38.9)	7 (30.4)		40 (47.1)	23 (45.1)	17 (50.0)	
1	33 (26.2)		16 (39.0)	8 (44.4)	8 (34.8)		17 (20.0)	9 (17.6)	8 (23.5)	
2	27 (21.4)		6 (14.6)	1 (5.6)	5 (21.7)		21 (24.7)	15 (29.4)	6 (17.6)	
3	7 (5.6)		2 (4.9)	1 (5.6)	1 (4.3)		5 (5.9)	4 (7.8)	1 (2.9)	
Unknown	5 (4.0)		3 (7.3)	1 (5.6)	2 (8.7)		2 (2.4)	0 (0)	2 (5.9)	
Margin (%)		0.335				0.851				0.730
Free	115 (91.3)		36 (87.8)	16 (88.9)	20 (87.0)		79 (92.9)	47 (92.2)	32 (94.1)	
Not free	11 (8.7)		5 (12.2)	2 (11.1)	3 (13.0)		6 (7.1)	4 (7.8)	2 (5.9)	

* *p* value comparing PF4500 group and PF5040 group; ** *p* value comparing TLND ≤ 17 and TLND > 17 in the PF4500 group; *** *p* value comparing TLND ≤ 17 and TLND > 17 in the PF5040 group; IQR: interquartile range; TLND: total lymph node dissection number; LND: lymph node dissection; PNI: perineural invasion; LVI: lymphovascular invasion; TRG: tumor regression grade (0: complete response; 1: near complete response; 2: partial response; 3: poor response).

**Table 2 jcm-11-05059-t002:** Univariate and multivariate Cox regression model for overall survival.

	Univariate			Multivariate		
Variables	HR	95% CI	*p* Value	HR	95% CI	*p* Value
Radical resection			0.011			0.696
R0	1			1		
R1	2.68	1.26–5.69		0.82	0.30–2.23	
pCR			<0.001			0.010
No	1			1		
Yes	0.31	0.18–0.55		0.44	0.24–0.82	
Tumor differentiation			<0.001			0.171
Well/moderate	1			1		
Poor	3.99	1.95–8.16		2.00	0.74–5.38	
PNI			<0.001			0.080
Negative	1			1		
Positive	3.78	2.05–6.98		2.18	0.91–5.23	
LVI			<0.001			0.702
Negative	1			1		
Positive	3.42	1.97–5.92		1.19	0.48–2.94	
Pathological stage			0.007			0.181
I/II/III	1			1		
IV	2.79	1.33–5.88		1.76	0.77–4.05	

Abbreviations: HR: hazard ratio; CI: confidence interval; pCR: pathological complete response; PNI: perineural invasion; LVI: lymphovascular invasion.

**Table 3 jcm-11-05059-t003:** Univariate and multivariate Cox regression model for disease-free survival.

	Univariate			Multivariate		
Variables	HR	95% CI	*p* Value	HR	95% CI	*p* Value
Radical resection			0.001			0.451
R0	1			1		
R1	3.10	1.62–5.93		1.44	0.56–3.69	
pCR			<0.001			0.035
No	1			1		
Yes	0.39	0.24–0.62		0.56	0.32–0.96	
Tumor differentiation			<0.001			0.968
Well/moderate	1			1		
Poor	3.65	1.65–6.84		1.02	0.37–2.79	
PNI			<0.001			0.417
Negative	1			1		
Positive	2.94	1.68–5.15		1.36	0.65–2.83	
LVI			<0.001			0.115
Negative	1			1		
Positive	3.21	1.95–5.28		1.82	0.86–3.84	
Pathological stage			<0.001			0.029
I/II/III	1			1		
IV	3.70	1.83–7.47		2.65	1.11–6.33	

Abbreviations: HR: hazard ratio; CI: confidence interval; pCR: pathological complete response; PNI: perineural invasion; LVI: lymphovascular invasion.

**Table 4 jcm-11-05059-t004:** Univariate and multivariate Cox regression model for locoregional-specific DFS.

	Univariate			Multivariate		
Variables	HR	95% CI	*p* Value	HR	95% CI	*p* Value
Radical resection			0.024			0.857
R0	1			1		
R1	2.37	1.12–5.00		0.91	0.33–2.53	
pCR			0.001			0.023
No	1			1		
Yes	0.42	0.25–0.69		0.53	0.30–0.92	
Tumor differentiation			0.004			0.336
Well/moderate	1			1		
Poor	2.83	1.40–5.74		1.60	0.61–4.19	
PNI			<0.001			0.091
Negative	1			1		
Positive	3.26	1.79–5.94		2.03	0.89–4.62	
LVI			<0.001			0.711
Negative	1			1		
Positive	2.79	1.64–4.76		1.17	0.51–2.70	

Abbreviations: DFS: disease-free survival; HR: hazard ratio; CI: confidence interval; pCR: pathological complete response; PNI: perineural invasion; LVI: lymphovascular invasion.

**Table 5 jcm-11-05059-t005:** Perioperative outcomes of patients in PF4500 and PF5040 group.

	PF4500, N = 41	PF5040, N = 85	*p* Value
Operative time (min; median, IQR)	540 (479–625)	471 (390–562)	0.001
Chyle leaks	2 (4.9%)	6 (7.1%)	0.638
Anastomotic leaks	6 (14.6%)	8 (9.4%)	0.381
Type II	6 (14.6%)	6 (7.1%)	0.473
Type III	0	2 (2.4%)	
Vocal cord palsy	12 (29.3%)	9 (10.6%)	0.008
Type I	6 (14.6%)	4 (4.7%)	0.801
Type II	6 (14.6%)	5 (5.9%)	
Pneumonia	5 (12.2%)	11 (12.9%)	0.906
Wound infection	2 (4.9%)	6 (7.1%)	0.638
LOS (days, median, IQR)	16 (13–26)	16 (13–25)	0.857
Complication grade ≥ III	11 (26.8%)	14 (16.5%)	0.172
30-day mortality	0	2 (2.4%)	0.322
90-day mortality	0	2 (2.4%)	0.322

Abbreviations: IQR: interquartile range; LOS: length of stay.

**Table 6 jcm-11-05059-t006:** Adverse events of neoadjuvant chemoradiotherapy of patients in PF4500 and PF5040 group.

	PF4500, N = 41		PF5040, N = 85		*p* Value *
	Grade 3 or 4	All Grade	Grade 3 or 4	All Grade	
Hematological					
Leukopenia	7 (17.1)	8 (19.5)	15 (17.6)	23 (27.1)	0.357
Anemia	2 (4.9)	4 (9.8)	3 (3.5)	10 (11.8)	0.737
Thrombocytopenia	1 (2.4)	1 (2.4)	2 (2.4)	8 (9.4)	0.269
Non-hematological					
Gastrointestinal tract **	1 (2.4)	17 (41.5)	3 (3.5)	46 (54.1)	0.183
Respiratory system ***	0 (0)	1 (2.4)	1 (1.2)	6 (7.1)	0.426

* *p* value comparing all grade adverse events in the PF4500 group and PF5040 group; ** adverse events of gastrointestinal tract included anorexia, esophagitis, mucositis, nausea, vomiting, and diarrhea; *** adverse events of respiratory system included pneumonitis, cough, dyspnea, and hiccups.

## Data Availability

Data presented in this study will be provided upon reasonable request.

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
