# Peer review of "Prognostic Effect of the Dose of Radiation Therapy and Extent of Lymphadenectomy in Patients Receiving Neoadjuvant Chemoradiotherapy for Esophageal Squamous Carcinoma"

_jcm, 2022, doi:10.3390/jcm11175059_

Round 1
Reviewer 1 Report
This is an interesting manuscript written by Chu-Pin Pai et al on the effects of neoadjuvant CTRT in the treatment of esophageal SCC.
The paper is well written and presents data based on adequate statistical investigation.
As this is a retrospective study, I advise the authors to clarify in the text the reasons why some patients received a radiation doses of 45.0 Gy and others of 50.4 Gy.
Moreover it would be interesting to report toxicity during the treatment or development of radio related pathologies derived from the different dose.
Author Response
We appreciate the reviewer’s comments and suggestions. Below are our answers to the comments.
Comment 1:
As this is a retrospective study, I advise the authors to clarify in the text the reasons why some patients received a radiation doses of 45.0 Gy and others of 50.4 Gy.
Answer 1: The decision of utilizing a radiation dose of 45.0 or 50.4 Gy was mainly made based on the updated evidences at the time of treatment, disease status and medical condition of the patient. To optimize the treatment planning, a thorough discussion in the multidisciplinary team conference was performed.
We would like to give a detailed explanation on this in the “methods” section in the “NCRT and surgery” paragraph as suggested by the reviewer 1. We thank the reviewer’s comments.
Changes 1: The above information has been added in the revised manuscript. (line 78-82)
Comment 2:
Moreover, it would be interesting to report toxicity during the treatment or development of radio related pathologies derived from the different dose.
Answer 2: We were able to identify similar rates of adverse events in both groups (summarized in Table 6). When looking specifically at adverse events grading more than 3, comparable results were also identified between the two groups. Moreover, no severe adverse events intercepting the neoadjuvant therapy were reported. We would like to add the above information into the manuscript. We thank the reviewer’s comment.
Changes 2: The above information summarized in Table 6 has been added in the revised manuscript. (lines 211-213)
Reviewer 2 Report
1. For lines 40-41, “Despite the wide range of acceptable RT doses, the doses most commonly used in Taiwan are 45.0 and 50.4 Gy”. The author should also explain other country or the world-wide used RT doses, not only mentioned Taiwan alone.
2. The authors should explain “The postoperative length of stay and unplanned re- admission within 30 days, which can serve as a surrogate for post-operative complications, were also similar between the groups. In our study, similar rates of R0 resection, overall complications with Clavien–Dindo grade ≥ III and short-term mortality were identified between the PF4500 and PF5040 groups. Overall, both regimens in our study should be considered as a safe and effective neoadjuvant treatment.” more detailed. (Lines 238-243)
3. The authors should also explain the sentences “In our study, when stratified by total LND number, the OS, DFS and locoregional-specific DFS were comparable between the PF4500 and PF5040 groups (Figure 2a, 2b, 3b). Nonetheless, the ypN0 and ypN downstaging rate was similar between the PF4500 and PF5040 groups.” more detailed. (Lines 251-254)
4. Figure 1(a) and 1(b) are asymmetric at “height” and “number at risk”. Please revise it. (Page 5)
Author Response
We appreciate the reviewer’s comments and suggestions. Below are our answers to the comments.
Comment 1:
For lines 40-41, “Despite the wide range of acceptable RT doses, the doses most commonly used in Taiwan are 45.0 and 50.4 Gy”. The author should also explain other country or the world-wide used RT doses, not only mentioned Taiwan alone.
Answer 1:
“Sensitizing chemotherapy with concurrent 41.4 Gy RT (CROSS) is the current standard of care for neoadjuvant chemoradiotherapy (nCRT) in Western regions, mainly in response to esophageal adenocarcinoma. In Asia, more than 90% of all esophageal cancer diagnosed are esophageal squamous cell carcinoma. Cisplatin plus 5-fluorouracil (PF)-based regimens with an RT dose between 41.4 and 50.4 Gy has proven effective in previous clinical trials and is frequently used in Asia.” We would like to give a more detailed explanation on the RT doses used in other regions according to the reviewer’s suggestion. We thank the reviewer’s comments.
Changes 1: The above information has been added in the revised manuscript. (lines 38-41)
Comment 2:
The authors should explain “The postoperative length of stay and unplanned re- admission within 30 days, which can serve as a surrogate for post-operative complications, were also similar between the groups. In our study, similar rates of R0 resection, overall complications with Clavien–Dindo grade ≥ III and short-term mortality were identified between the PF4500 and PF5040 groups. Overall, both regimens in our study should be considered as a safe and effective neoadjuvant treatment.” more detailed. (Lines 238-243)
Answer 2:
“In our study, similar rates of neoadjuvant therapy related adverse events were identified between the PF4500 and PF5040 groups. Regarding the postoperative course, rates of overall complications with Clavien–Dindo grade ≥ III, postoperative length of stay, unplanned readmission within 30 days and short-term mortality were also comparable between the two groups. The PF5040 group, despite using a higher RT dose, exhibited similar results of treatment toxicity compared to the PF4500 group. Overall, both regimens in our study should be considered as a safe and effective neoadjuvant treatment.”
In line 235 to 239, we first described the comparable results of tumor control and response of the two groups. Then, we illustrated the treatment related adverse events and perioperative outcomes (morbidities and mortalities). We tried to compare the difference of the two regimens, in terms of safety and effectiveness. We would like to explain the above information in detail within the manuscript. We thank the reviewer’s comments.
Changes 2: In the revised manuscript, the above information has been added and the paragraph has been rearranged to a more readable fashion. (lines 253-260)
Comment 3:
The authors should also explain the sentences “In our study, when stratified by total LND number, the OS, DFS and locoregional-specific DFS were comparable between the PF4500 and PF5040 groups (Figure 2a, 2b, 3b). Nonetheless, the ypN0 and ypN downstaging rate was similar between the PF4500 and PF5040 groups.” more detailed. (Lines 251-254)
Answer 3:
“In our study, when stratified by total LND number, the OS, DFS and locoregional-specific DFS were comparable between the PF4500 and PF5040 groups (Figure 2a, 2b, 3b). Regarding loco-regional control, a more aggressive LND did not translate into a significantly improved outcome. On the other hand, despite using a higher RT dose in the PF5040 group, the ypN0 and ypN downstaging rate was similar comparing to the PF4500 group. In terms of loco-regional control of lymph node metastasis, no significant superiority was observed in the PF5040 group over the PF4500 group, and a disfavoring outcome could occur once the LND was compromised. Therefore, despite no additional benefits being detected, LND should not be altered between the two regimens.” We would like to explain the concept in more detail. We thank the Reviewer`s comment.
Changes 3: The above information has been added in the revised manuscript. (lines 271-279)
Comment 4:
Figure 1(a) and 1(b) are asymmetric at “height” and “number at risk”. Please revise it. (Page 5)
Answer 4:
We adjusted the asymmetricity of the Figure 1(a) and 1(b). We thank the Reviewer`s comment.
Changes 4: The adjustment has been done in the revised manuscript.